# Emotion regulation and compassion fatigue in mental health professionals in a context of stress: A longitudinal study

P. Brillon[1], M. Dewar[2,3], V. Lapointe[1], A. Paradis[1], F. L. Philippe[1]*

1 Department of Psychology, University of Quebec at Montreal, Montreal, Canada, 2 Institut du Savoir Montfort, Ottawa, Canada, 3 School of Psychology, Ottawa University, Ottawa, Canada

* philippe.frederick@uqam.ca

## Abstract

### Introduction

Mental health professionals (MHP) are exposed to several stressors and have emotionally demanding jobs. They must effectively manage their emotions within their everyday practice. Emotion regulation is therefore a key element in understanding how MHPs can protect themselves psychologically. Abundant research shows that healthy and effective emotion regulation can protect against the negative impact of stress on compassion fatigue. However, this perspective does not consider the dynamic interaction that emotion regulation and compassion fatigue can have over time. A much less researched perspective is how compassion fatigue can change emotion regulation styles over time. The present research focused on this dynamic perspective.

### Objective

We took advantage of the COVID-19 pandemic, a stressful period for MHPs, to study the effect of perceived stress on the direction of the changes in emotion regulation styles and compassion fatigue over time.

### Methods

Data on stressors, perceived stress, emotion regulation styles (i.e., dysregulation, integration, and suppression), and compassion fatigue were collected from 390 MHPs at two time points over ten months.

### Results

Findings from a cross-lagged path analysis suggests that perceived stress predicted increases in dysregulation over time. Moreover, there were bidirectional longitudinal associations between dysregulation and compassion fatigue, with each predicting increases in compassion fatigue and dysregulation over time, respectively.

**Data Availability Statement:** All relevant data for this study are publicly available from the OSF repository (https://osf.io/26txq/).

**Funding:** This work was supported by a Strategic Research Chair by the University of Quebec at Montreal (20210101 to FLP). The funder had no role in study design, data collection and analysis, decision to publish, or preparation of the manuscript.

**Competing interests:** The authors have declared that no competing interests exist.

## Conclusions

This study contributes to the limited research on the factors that influence how MHPs regulate their emotions and their susceptibility to compassion fatigue. Implications of emotion regulation for MHPs' own mental health and ability to do their work effectively are discussed.

## Introduction

Psychologists, social workers, mental health nurses, psychiatrists, and other mental health professionals (MHPs) provide care to promote the mental wellbeing of vulnerable or distressed populations. In doing so, MHPs are exposed firsthand to patients' intense emotions, life stories, and difficulties. These professionals encounter several stressors. Some are intrinsic to healthcare professions, such as working "face-to-face" with other people and having emotionally charged interactions [1]. Other stressors include a high workload, limited resources to attend to the needs of patients, impostor syndrome, or poor coordination of services [2]. Because of the COVID-19 pandemic, healthcare workers have had to cope with additional challenges such as disruption of services, temporary reassignments to different roles, transition to telehealth, more difficulty balancing professional and personal life, increased isolation, the threat of the COVID-19 virus, increased distress in service users, and higher demands for mental health care [3–6]. MHPs are known to have great psychological resources; they tend to display stronger coping capacities and resilience compared to other professionals [7]. Nonetheless, the stressors they face increase their risk of compassion fatigue [8].

Compassion fatigue describes a diminished ability to bear another's suffering that can result from prolonged caregiving and regular contact with psychological distress [9]. Hence, MHPs face an especially high risk of compassion fatigue [8]. Compassion fatigue tends to manifest itself in exhaustion, frustration, anger, and a diminished ability to feel empathy for others [9,10]. These negative feelings are accompanied by a depleted ability to cope with common work tasks and by the perception that efforts made at work are "pointless" or make little difference [9,10]. At any given time, between 7.7% and 28.6% of professional groups (e.g., psychologists, social workers) report high levels of compassion fatigue [8].

Several studies suggest that compassion fatigue can significantly deteriorate the quality of psychological care offered to patients [11,12]. In mitigating these issues, emotion regulation naturally emerges as a key concept in understanding how MHPs can protect themselves from the stress of their professions and maintain good psychological adjustment.

### Emotion regulation among mental health professionals

Managing emotions experienced during interactions with patients is one of the most important but challenging aspects of mental health work [13]. Indeed, providing compassionate care requires that MHPs remain calm, open-minded, and empathetic despite the stressors experienced. To maintain this good professional posture, MHPs must become aware of, manage, delay, or alter the emotions they feel, reveal, and express [14]. Processing emotions is key to maintaining a healthy boundary between professional and personal life, helping MHPs avoid bringing patients' distress home or starting their workday already emotionally overwhelmed [8].

This process is known as emotion regulation [15,16]. Emotion regulation is defined as an "attempt to influence which emotions one has, when one experiences them, for how long, in

what intensity, and how one expresses them" [17,18 p. 200]. Emotions, whether pleasant or unpleasant, serve as a source of important information to promote personal growth, advocate for needs, and foster well-being [15,16,19]. In fact, in the context of their work, MHPs use their emotions to form clinical impressions and guide their interventions; during interactions with patients, they carefully choose to share some thoughts and feelings and hide others to the benefit of patients [20]. In doing so, they display strong emotion regulation capacities.

Based on recent theoretical models of emotion regulation strategies [e.g., 21] and meta-analyses [e.g., 22], self-determination theory of emotion regulation suggests that there are three styles of emotion regulation; (1) integrative regulation, (2) suppressive regulation, and (3) emotional dysregulation [18,19].

Integrative emotion regulation involves being mindful and receptive to internal emotional experiences using a non-judgmental, benevolent, and curious stance. Emotions are seen as important sources of information about values, personal goals and boundaries [18]. This style is considered the most adaptive form of regulation because it allows for self-awareness, value- and goal-oriented actions, and effective expression of emotions [18,23]. For its part, suppressive regulation involves trying to minimize, avoid, deny, hide, change, or ignore emotions that are experienced as pressuring, frightening, or dangerous [15,18]. Because negative emotions remain unattended or unexpressed, suppression tends to maintain or increase emotions over time, and is therefore considered less adaptive [19,23]. Lastly, dysregulation style designates a mode when emotions are experienced as overwhelming or disorganizing; they cannot be managed and significantly interfere with everyday functioning and interpersonal relationships [15,18]. This may lead people to act impulsively, withdraw, self-harm, or have emotional outbursts. Difficulties in emotion regulation (i.e., not accepting emotions, not approaching them with adaptive actions, or having impulsive reactions) are among the well-defined transdiagnostic mechanisms of psychopathology [see for reviews 24,25].

### Emotion regulation and compassion fatigue: A bidirectional relationship?

Much of the research on emotion regulation has examined how regulation capacities are used to mitigate the effect of stress on psychological adjustment [25,26]. This line of research has been specifically focused on how emotion regulation styles can help reduce psychological symptoms [e.g., 27], like compassion fatigue [e.g., 28]. Through this lens, maladaptive emotion regulation styles can exacerbate psychological symptoms in times of stress, whereas more adaptative styles can reduce them.

A line of research that remains to be investigated is how this relationship can be bidirectional and therefore how psychological symptoms can actually change emotion regulation styles [29,30]. While emotion regulation is often conceptualized as a stable trait, more recent research shows that it is context-specific and that it can change over time [31]. As such, much remains to be understood about the factors that contribute to changes in emotion regulation styles over time [32,33]. To our knowledge, this question has never been examined in MHPs or in relation to compassion fatigue. Yet, understanding the changes over time in the emotion regulation styles adopted by MHPs according to their compassion fatigue is an essential research objective considering the importance of emotional regulation in these professionals' well-being and effectiveness.

### Objectives of the current study

COVID-19 has been a stressful period for MHPs [3–6] and, therefore, provided a well-suited context to study how emotion regulation styles may change under stress. More specifically, we aimed to understand how stressors and perceived stress related to COVID-19 might be

associated with changes in emotion regulation styles over time and contribute to compassion fatigue in MHPs during a stressful period. Moreover, we will examine the reverse relationship, that is, whether compassion fatigue in MHPs can contribute to change their emotion regulation styles over time.

We hypothesized that COVID-related stressors (i.e., physical and contextual) would be associated with more perceived stress during the COVID-19 period. Perceived stress was expected to predict increases in maladaptive emotion regulation styles (i.e., suppression and dysregulation) and decreases in adaptative styles (i.e., integration) and increases in compassion fatigue over time. In addition, in line with the classical literature on emotion regulation and psychological adjustment, we expected that maladaptive emotion regulation styles at T1 would predict increases in compassion fatigue at T2, whereas adaptive regulation styles at T1 would predict decreases in compassion fatigue T2. In line with more recent literature examining how emotion regulation styles can change over time [29,31], we also expected compassion fatigue at T1 to predict changes in emotion regulation styles at T2, by increasing maladaptive emotion regulation styles and decreasing adaptive styles. All these relationships will be examined while controlling for baseline level of both emotion regulation and compassion fatigue. The model was also adjusted for variables relevant to MHPs and the context of the COVID pandemic, that is, age, sex, MHPs' workload, and work modalities (e.g., telepractice). Sensitivity analyses were also conducted by examining whether MHPs' workload or work modalities would moderate the results.

## Method

### Participants and procedure

The sample was comprised of 390 MHPs who completed both data collection points. MHPs were between 21 and 72 years of age ($M$ = 42.34, $SD$ = 11.34), were mostly highly educated (over 90% had at least one university degree) and identified in large part as women (89.5%). Several types of MHPs participated: most were psychologists (38.7%), psychosocial counselors (16.7%), social workers (11.3%), and psychoeducators (9.5%). The remainder of the sample included mental health nurses, psychiatrists, criminologists, crisis counselors, sexologists, art therapists, and various other types of professionals. MHPs mostly worked in various settings: 39.5% in the public sector (e.g., hospitals, youth protection), 28.7% in community organizations, 25.4% in private practice, and 6.4% in other settings. The majority worked with adult populations (78.2%) and with children and adolescents (17.7%), but 3.3% specified working with older adults and 0.8% with professionals or organizations. As the study took place during the COVID-19 pandemic, we inquired about the participants' current work modality: 15.9% worked exclusively in person, 59.2% exclusively using telepractice, and 23.5% combined the two modalities (the remaining 1.4% indicated that this question did not apply).

MHPs were recruited through professional associations which agreed to send an email invitation to the study (provided by the research team) to their members. Participants were informed that the study was confidential and included two time points. Inclusion criteria were to be 20 years of age or older, to be working as a MHPs during the study. Those who agreed to participate provided consent and completed an online survey on a private secure web platform. As compensation for their time, participants were entered into a draw for one of three prizes of $125.

Ten months separated both data collection points: Time 1 (T1) took place from May 1st to July 20th, 2020, and Time 2 (T2) spanned from March 11th to June 6th, 2021. Although it was hard to predict at that time, we were successful in recruiting participants at both phases at the end of covid waves as they occurred in Quebec, Canada—that is, the end of Wave 1 for Time 1

(Wave 1 spanned from February 25[th], 2020 to July 11[th], 2020) and the end of Wave 2 for Time (Wave 2 spanned from August 23th, 2021 to March 20[th], 2021). Both waves were characterized by approximately the same number of deaths, that is, 5,687 deaths for Wave 1 and 4,904 deaths for Wave 2 [34], thus making the two phases of our study approximately comparable. Moreover, these two waves were the ones characterized by the greatest number of deaths over the whole pandemic in Quebec, Canada. The response rate at T2 was 47% from the initial sample of 839 MHPs. To assess dropout bias, all T1 study variables were entered as independent variables in a logistic regression with dropout status as the dependent variable. Results revealed that no variable was significantly associated with dropout status at $p < .05$. Ethics approval was granted by the Ethics Committee of the University of Quebec at Montreal (2021–1756, 2070). Participants provided online consent by checking a box confirming that they have read the consent form and voluntarily consent to participate in the study.

## Measures

**Sociodemographic and clinical practice characteristics.**   Questions enquired about sociodemographic characteristics (e.g., age, sex), and about the participants' clinical practice (e.g., work modality [in person = 0, telepractice = 1] and change in workload [same or reduced = 0, increased = 1]). These variables will be used as covariates in the main model and work modality and workload will also be examined as moderators of the results.

**Physical stressors.**   Participants were asked to select any current health conditions from a list of conditions which have been linked to increased risks for COVID-19 complications (e.g., hypertension, heart disease, immune system diseases, diabetes, obesity, cancer). The number of conditions selected served as a total score (possible range: 0 to 13).

**Contextual stressors.**   Participants were asked about any changes that they had experienced in the context of the COVID-19 pandemic (e.g., reduced income, children out of school). The total number of stressors was used (possible range: 0 to 13).

**Perceived stress during the COVID period.**   The degree to which participants had health or safety fears in the context of COVID-19 was assessed using a scale validated using a general population sample and recruited during the pandemic [35]. This scale included six items (e.g., "I am afraid for the life of my loved ones", "I am afraid for my own health") rated on a 7-point Likert scale (1 = Strongly disagree, 7 = Strongly agree).

**Emotion regulation strategies.**   Three subscales of an adapted version of the Emotion Regulation Scale [19,29] measured three types of emotional regulation strategies; *dysregulation*, *integration* and *suppression* using a 7-point Likert scale (1 = Strongly disagree, 7 = Strongly agree). Six items measured dysregulation (e.g., "When I experience emotions, I usually feel that I have little control over my behavior"), seven items measured integration (e.g., "When I feel negative emotions, I usually try to understand the reasons why"), and seven items focused on suppression (e.g., "I try to ignore my negative emotions").

**Compassion fatigue.**   A subscale of the Professional Quality of Life Scale [10] inquired about the frequency to which MHPs experienced signs of compassion fatigue (e.g., "I feel worn out because of my work as a mental health professional."). Each of the 10 items are rated to on a 5-point Likert scale (1 = Never, 5 = Very often).

All measures demonstrated satisfactory internal consistency (see Table 1 for Cronbach's alpha values).

## Statistical analysis

Descriptive statistics, correlational analysis, and Cronbach's α for each of the measures were calculated and presented in S1 Table. A regression analysis was then conducted to examine the

**Table 1. Standardized path coefficients of the cross-lagged model with control variables.**

| Dependent Variables | α | Perceived stress | | | Age | | | Sex | | | Work modalities | | | Workload | | |
|---|---|---|---|---|---|---|---|---|---|---|---|---|---|---|---|---|
| | | β | SE | z | β | SE | z | β | SE | z | β | SE | z | β | SE | z |
| Dysregulation T1 | .80 | 0.256 | 0.046 | **5.528** | -0.162 | 0.048 | **-3.363** | -0.079 | 0.049 | -1.628 | -0.017 | 0.048 | -0.361 | 0.041 | 0.048 | 0.849 |
| Integration T1 | .85 | -0.048 | 0.050 | -0.967 | 0.069 | 0.051 | 1.357 | -0.164 | 0.050 | **-3.285** | 0.086 | 0.050 | 1.717 | 0.058 | 0.050 | 1.153 |
| Suppression T1 | .90 | 0.049 | 0.049 | 0.998 | -0.114 | 0.050 | **-2.259** | 0.084 | 0.050 | 1.665 | -0.153 | 0.049 | **-3.100** | 0.002 | 0.050 | 0.040 |
| Compassion fatigue T1 | .79 | 0.171 | 0.048 | **3.580** | 0.004 | 0.049 | 0.087 | -0.127 | 0.049 | **-2.587** | -0.116 | 0.049 | **-2.383** | 0.162 | 0.048 | **3.358** |
| Dysregulation T2 | .79 | 0.082 | 0.039 | **2.123** | -0.104 | 0.039 | **-2.676** | -0.021 | 0.039 | -0.532 | 0.037 | 0.038 | 0.974 | -0.052 | 0.038 | -1.367 |
| Integration T2 | .86 | 0.052 | 0.043 | 1.222 | 0.013 | 0.043 | 0.299 | -0.059 | 0.043 | -1.380 | 0.012 | 0.042 | 0.288 | -0.073 | 0.042 | -1.747 |
| Suppression T2 | .88 | 0.054 | 0.042 | 1.282 | -0.001 | 0.042 | -0.024 | -0.029 | 0.043 | -0.689 | -0.054 | 0.042 | -1.281 | 0.052 | 0.042 | 1.245 |
| Compassion fatigue T2 | .80 | 0.036 | 0.042 | 0.847 | -0.009 | 0.042 | -0.204 | -0.010 | 0.043 | -0.235 | 0.016 | 0.042 | 0.389 | -0.050 | 0.042 | -1.195 |

| Dependent Variables | Dysregulation T1 | | | Integration T1 | | | Suppression T1 | | | Compassion fatigue T1 | | |
|---|---|---|---|---|---|---|---|---|---|---|---|---|
| | β | SE | z | β | SE | z | β | SE | z | β | SE | z |
| Dysregulation T2 | 0.581 | 0.037 | **15.653** | -0.036 | 0.039 | -0.914 | 0.003 | 0.041 | 0.084 | 0.088 | 0.042 | **2.129** |
| Integration T2 | -0.049 | 0.047 | -1.040 | 0.497 | 0.038 | **12.975** | -0.153 | 0.044 | **-3.441** | -0.067 | 0.046 | -1.480 |
| Suppression T2 | -0.026 | 0.047 | -0.554 | -0.055 | 0.043 | -1.284 | 0.564 | 0.038 | **14.767** | 0.015 | 0.045 | 0.321 |
| Compassion fatigue T2 | 0.120 | 0.046 | **2.579** | -0.017 | 0.043 | -0.389 | -0.028 | 0.044 | -0.633 | 0.545 | 0.040 | **13.693** |

*Note.* T1 = Time 1; T2 = Time 2; α = Cronbach's Alpha. Z test scores in bold are $p < .05$.

stressors associated with perceived stress. A cross-lagged path analysis model was then conducted with Mplus 7.11 to examine the association of perceived stress and emotion regulation and compassion fatigue over time. Moreover, the cross-lagged model examined the bidirectional relationship between emotion regulation and compassion fatigue over time. Such a model therefore allowed us to examine how perceived stress, emotion regulation and compassion fatigue could predict changes over time in emotion regulation and compassion fatigue. In such a model, change correspond to a residualized change score and not change in the intercept of the variables. This model was adjusted for sex, age, work modalities, and changes in workload.

Sensitivity analyses were also conducted to determine whether work modalities and workload would moderate any of the significant associations found. For each moderator, two models were constructed. In the first, all possible associations were included, and all parameters freely estimated; this saturated model with zero degree of freedom produces perfect fit indices. In the second model, the main significant associations found in the cross-lagged path analysis were constrained to be equal across the two levels of the given moderator, producing fit indices. With the two sets of fit indices, those of the saturated and the constrained models, we were able to examine whether there was a moderation effect by comparing the fit indices. When the fit indices of the saturated and constrained models were considered equal, this suggested an absence of a moderation effect. To be of equal fit, norms recommend that models have a non-statistically significant chi-square and that they differ by no more than 0.010 on the Comparative Fit Index (CFI) and Tucker-Lewis Index (TLI), and no more than 0.015 on the Root-Mean-Square Error of Approximation (RMSEA) [36,37].

## Results

Correlational results showed that stressors measured at T1 (i.e., physical and contextual) were both positively associated with perceived stress at T1 (see S1 Table). Moreover, a linear regression analysis showed that both physical ($\beta = .21$, $b = .51$, $SE = .12$, $p < .001$) and contextual stressors ($\beta = .16$, $b = .16$, $SE = .05$, $p < .001$) were both independently contributing to

perceived stress. These findings suggest that the COVID-19 pandemic did increase the perceived stress of MHPs.

We then turned to examine whether this perceived stress would lead to change in emotion regulation (i.e., dysregulation, integration, suppression) and compassion fatigue over time. We also investigated whether emotion regulation would lead to changes in compassion fatigue over time and/or whether compassion fatigue would lead to changes in the use of emotion regulation styles over time. All standardized path coefficients obtained in this saturated model, including those of the control variables, are presented in Table 1.

All auto-regressive paths were significant; all T1 emotional regulation styles and compassion fatigue were strongly associated with their T2 measure. Results showed that perceived stress was cross-sectionally associated with dysregulation and predicted a longitudinal increase in dysregulation over. However, it was not associated with integration or suppression styles either cross-sectionally or longitudinally. Perceived stress was cross-sectionally positively associated with compassion fatigue, but it did not predict a longitudinal increase in compassion fatigue over time. Results probing the bidirectional relationship between emotion regulation and compassion fatigue partly supported our hypotheses. Dysregulation predicted an increase in compassion fatigue over time, but the suppression and integrative styles were not associated with compassion fatigue. Moreover, compassion fatigue predicted an increase in dysregulation over time, but no change in integrative and suppression styles. These results highlight the feedback loop occurring over time between dysregulation and compassion fatigue, as triggered by perceived stress (see Fig 1).

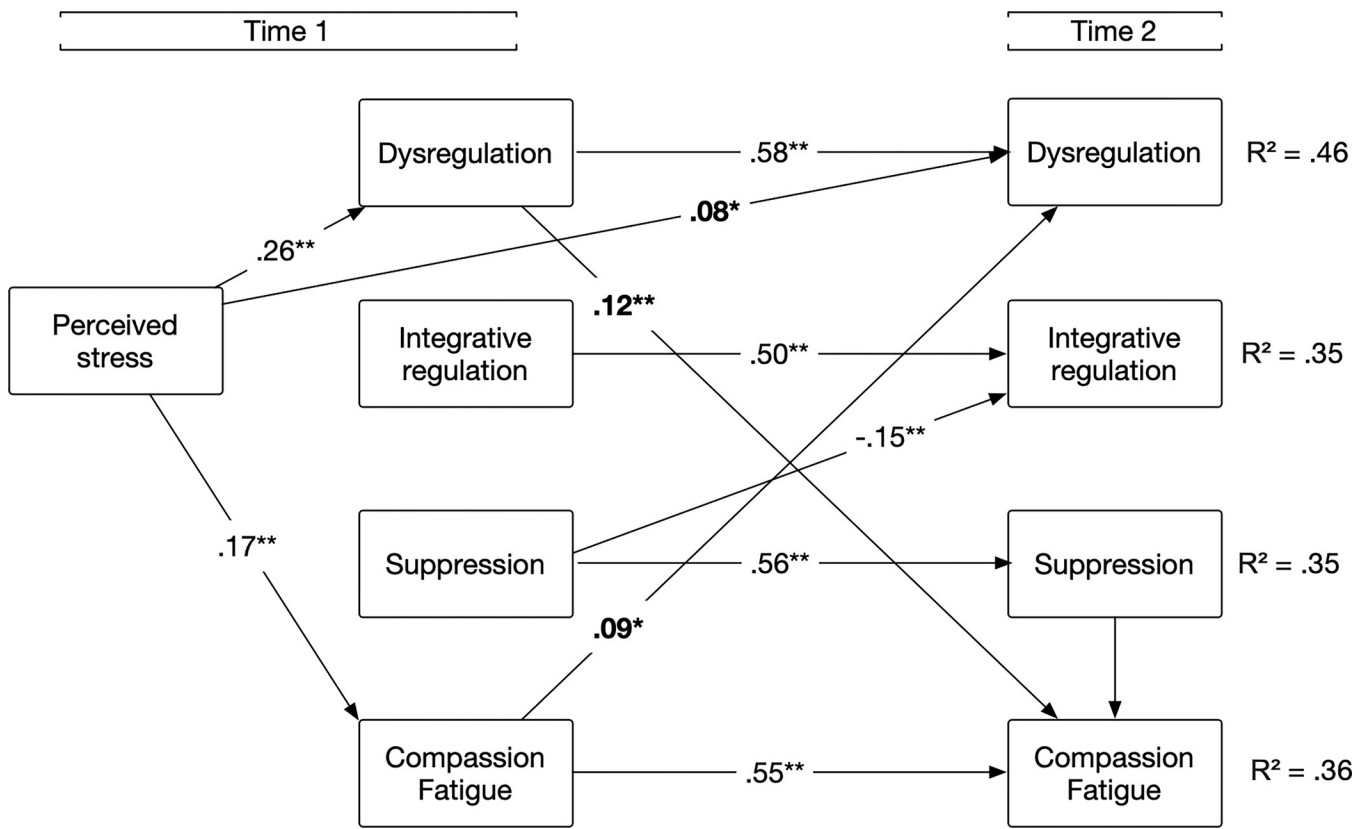

**Fig 1. Cross-lagged path analysis model of emotion regulation and compassion fatigue with perceived stress.** *Note.* All estimates are standardized. In the model, the covariations among measurement errors and the paths from the control variables were estimated but are not shown for the sake of clarity. Full parameter coefficients, including those from the control variables, can be found in Table 1. *$p < .05$, **$p < .01$.

Among the control variables, only age negatively predicted dysregulation at T2; specifically, being older predicted a decrease in dysregulation ($\beta$ = -.10, $SE$ = .04, $p$ = .006). Overall, this final model explained 46% of dysregulation, 35% of suppression, 35% of integration, and 36% of compassion fatigue at T2 ($p$ values < .001).

## Sensitivity analyses

We examined whether the above results would change as a function of MHPs' work modalities and changes in workload. Constraining all significant paths of the main model to be equal across the two levels of each moderator, no moderation effect was found, as suggested by the non-statistically significant chi-squares and nearly identical fit indices to those of the saturated model (see S2 Table for detailed results).

## Discussion

Providing psychological care requires that MHPs adequately manage their emotions to respond efficiently to both the demands of patients and their own needs [20]. This is a critical skill to prevent compassion fatigue. This study offers new insights into the current understanding of MHPs' emotion regulation capacities, specifically during times of personal and social adversity such as the COVID-19 pandemic. To our knowledge, this is the first study to examine how emotion regulation and compassion fatigue in MHPs influence each other over time across a stressful context.

### Stressors contribute to increased dysregulation

To date, few studies have tried to identify the contextual or motivational factors that influence why and how individuals regulate their emotions [21]. As hypothesized, findings indicated that stressors were associated with more perceived stress, which longitudinally predicted changes in emotional regulation styles, specifically increased dysregulation. Hence, our study points to stressors as a significant factor in the development and reinforcement of emotion dysregulation. This finding is significant because there is abundant literature that points to the critical role of emotion dysregulation in the development of psychopathology [26]. Emotion dysregulation is described as an overwhelming and disorganizing experience [18,38] that likely interferes with attention, decision-making, and empathy during clinical work. Thus, dysregulation appears to be the most detrimental emotional regulation style while providing psychological care.

Contrary to our hypotheses, we found that perceived stress was not significantly predictive of changes in integrative or suppressive emotion regulation styles over time. One explanation for this might have to do with the fact that MHPs use a combination of emotional regulation styles at once [39]. In the context of their work, MHPs likely integrate some emotions to the best of their capabilities, while concurrently minimizing or shielding themselves from a certain amount of emotion. However, in face of chronic or intense stressors, emotions could become overwhelming and, thus, only lead to an increase in emotional dysregulation rather than in integrative regulation or suppression—as these would already be consistently used. Therefore, over time, chronic stress could reinforce emotional dysregulation style. In line with this explanation, studies have shown that stress often initially triggers emotion dysregulation, which in turn disorganizes functioning and generates psychological symptoms [e.g., 29]. Suppression is then used as a strategy to reduce this overwhelming emotional state. This is especially evident after a traumatic event, where a high and disruptive arousal is initially experienced, followed by attempts from the individual to ignore the traumatic event and its associated arousing emotion and related triggers [40]. Future studies should examine how MHPs combine multiple

emotional regulation styles during their interactions with patients as this would allow us to better understand how emotional regulation styles influence each other and might reinforce one another.

Sensitivity analyses also showed that these findings did not differ as a function of work modality or workload. This suggests that the effect of perceived stress on dysregulation is robust and independent of changes in the work context of MHPs.

## Emotion regulation styles and compassion fatigue

All three emotion regulation styles were significantly correlated with compassion fatigue at both Times 1 and 2, with dysregulation and suppression positively associated with compassion fatigue and integrative regulation negatively associated with compassion fatigue, but only at Time 2. This aligns with previous research which shows that suppression and dysregulation tend to predict poorer psychological adjustment whereas integration tends to predict better adjustment [see meta-analysis, 41]. However, only dysregulation was significantly associated with more compassion fatigue over time. Therefore, dysregulation appears as a key psychological mechanism facilitating the development of compassion fatigue [e.g., 13].

At the same time, it was not surprising to find that dysregulation was associated with increased compassion fatigue. The role of emotional dysregulation in depression is well documented, as researchers have frequently highlighted that dysregulation impedes functioning, interpersonal relationships, and the capacity to increase, maintain, or decrease unpleasant and pleasant emotions over time [15,41]. In fact, when someone experiences emotional dysregulation, they may feel at the mercy of their emotions, feel overwhelmed by them, or express their emotions in a clumsy, unconstructive, or unwarranted way [15,18]. However, the present findings highlight the key role that dysregulation can play in MHPs compassion fatigue. This may mean that, while dysregulated, MHPs might be unable to keep a healthy distance from their own emotions, to be emotionally attuned to their patients and may make poorer therapeutic decisions. As a result, MHPs may feel (or actually be) less efficient, feel guilty, or display impostor syndrome, thereby contributing to the development of compassion fatigue [10,42].

One perhaps more surprising result in regard of the current literature is that compassion fatigue was found to change emotion regulation styles. More specifically experiencing compassion fatigue at Time 1 led to an increased use of dysregulation at Time 2. This finding is in line with the most recent literature which does not uniquely conceptualize emotion regulation as a stable trait, but also as a process that can change as a function of context and over time [29,31]. Specifically, the present research identified that perceived stress was a factor of changes in dysregulation over time and that the experience of mental symptoms such as compassion fatigue was a second independent process through which dysregulation can be increased over time. Taken together, these findings suggest that dysregulation and mental symptoms could mutually reinforce each other in an upward spiral, leading to a continual decline in functioning over time.

## Stability of emotion regulation styles

In our model, dysregulation, integration, and suppression were assessed at two time points, which allowed us to examine the stability of emotion regulation styles over time. As suggested by the abundant literature, emotion regulation is both dispositional and state-dependent [15,18]; we found that all three emotion regulation styles showed relative stability over the course of our study ($\beta$'s between .50 and .58), while also being state-dependent, influenced by perceived stress and compassion fatigue. One interesting finding showed that suppression at T1 predicted less emotional integration at T2. This suggests that, in addition to leading to

emotional build up, if suppressed emotions remain unattended, over time and under stress, this is associated with a reduction in one's capacity to integrate emotions in a healthy way. This suggests that while suppression does not appear to be altered by stressors, it is detrimental to MHPs over time, and, thus, MHPs should be cautious of emotional suppression.

## Clinical implications

The pandemic has highlighted the central role that MHPs play in treating and monitoring the mental health of the population and has consequently raised questions about their own emotion regulation and mental health. The extent to which MHPs can adapt to stressors and manage their emotions sustainably, the better their mental health, and the better their ability to treat and help their patients [e.g., 11].

Our study provides essential insight into how stressors contribute to more dysregulation and compassion fatigue in MHPs. Thus, our study implies that eliminating or reducing some of the stressors that have resulted from the COVID-19 pandemic would be beneficial for MHPs. For instance, we asked MHPs about stressors such as financial stress, the availability of childcare and difficulties with work-life balance. Therefore, one obvious strategy to directly reduce the stressors that MHPs face might be to revisit their work conditions. Other avenues could be to limit the number of complex cases taken on, increase professional support, team support, access to regular supervision within the workplace, and ensuring that that MHPs are well trained and comfortable with their mandate, as these have been identified in previous research as stressors [2,35]. Increasing governmental funding for mental health services, particularly within the public sector, also appears essential. Such funding would not only enhance access to mental health care for citizens, improving overall public well-being, but it would also support the mental health of MHPs by reducing their workload, providing better resources, and enabling more sustainable working conditions.

Our results also emphasize just how important it is to help MHPs develop and maintain strong emotion regulation capacities, namely, to reduce dysregulation. We encourage MHPs to reflect on their emotional regulation styles and on how they tend to feel, think, and act when they, for example, start feeling dysregulated. Our results can also encourage MHPs and the organizations that employ them to create space to promote healthy emotional integration, such as supervision sessions, reflective practice coaching, discussion forums, and workshops to popularize and provide information about emotional regulation styles. Other avenues include emotion regulation training, increasing mindfulness, self-awareness and self-care strategies as well as self-compassion and a caring attitude towards oneself could empower MHPs to take action to reduce dysregulation and suppression [e.g., 42–44]. Finally, financial and organizational arrangements to facilitate psychotherapy sessions for MHPs themselves could also be considered.

## Limitations and future research

Several limitations and avenues for future research are worth noting. First, although using a longitudinal design provided data on the direction of associations between stressors and emotional regulation strategies, this design did not allow for the investigation of causality. Moreover, the present study did not control for neuroticism, which could affect both dysregulation and compassion fatigue over time. Several other factors might have influenced the emotion regulation styles that MHPs chose to adopt and their subsequent psychological adjustment. Obviously, it would not be ethically possible to expose MHPs to stressors as intense as the COVID-19 outbreak. Yet, it could be fruitful in future studies to investigate how other factors, such as self-care strategies, flexible coping, resilience skills or self-compassion attitude towards

oneself contribute to how MHPs regulate their emotions. Also of note is that the present results were obtained during the COVID-19 pandemic. While it is reasonable to believe that similar results could be obtained under other stressful contexts, it remains unknown if they could replicate given the specificity of the context of the pandemic. Second, the present study only had two measurement time points, preventing us from examining changes in intercept and slope separately, which would have clarified the notion of change over time [e.g., 45]. Third, future studies should aim to further refine our findings. Our measure of emotion regulation assessed how MHPs regulate their emotions overall, but not specifically in the context of their work. Insofar as the adaptiveness of emotion regulation styles is largely context-dependent [31], we believe that an important next step in research would be to use measures directly linked to the work context or daily diary methods to understand in a more detailed way how stressors influence MHPs' emotion regulation styles, specifically in the professional context. Fourth, a significant proportion of MHPs did not complete the second assessment and it is impossible to determine the precise reasons for these non-completions. However, the final sample remained large, and no significant differences were found between those who completed only one or both assessment time points. This suggests that our results remain representative. Fifth, while efforts were put into place to recruit all types of MHPs, not all professions were equally represented. Thus, our results should be generalized to some professionals with caution. Finally, we used self-report measures for which bias is inevitable. Future studies could favor clinician assessments or objective measurements of stress biomarkers (e.g., cortisol levels). Despite these limitations, this study sheds new light on the emotion regulation capacities of MHPs and has important implications for the delivery of adequate psychological care and MHPs' well-being.

## Supporting information

**S1 Table. Correlation matrix of study variables.**
(DOCX)

**S2 Table. Fit of the multigroup analyses for the two moderators.**
(DOCX)

## Author Contributions

**Conceptualization:** P. Brillon, M. Dewar, A. Paradis, F. L. Philippe.

**Data curation:** F. L. Philippe.

**Formal analysis:** M. Dewar, V. Lapointe, A. Paradis, F. L. Philippe.

**Funding acquisition:** F. L. Philippe.

**Investigation:** F. L. Philippe.

**Methodology:** P. Brillon, M. Dewar, V. Lapointe, A. Paradis, F. L. Philippe.

**Project administration:** P. Brillon, F. L. Philippe.

**Supervision:** P. Brillon, A. Paradis, F. L. Philippe.

**Writing – original draft:** P. Brillon, M. Dewar, V. Lapointe, F. L. Philippe.

**Writing – review & editing:** P. Brillon, M. Dewar, V. Lapointe, F. L. Philippe.

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
