## [Decision Letter · Decision Letter 0]

22 Aug 2024

PMEN-D-24-00323

Stressors Change Emotion Regulation in Mental Health Professionals Over Time: Consequences for Depression and Compassion Fatigue

PLOS Mental Health

Dear Dr. Philippe,

Thank you for submitting your manuscript to PLOS Mental Health. After careful consideration, we feel that it has merit but does not fully meet PLOS Mental Health’s publication criteria as it currently stands. Therefore, we invite you to submit a revised version of the manuscript that addresses the points raised during the review process.

Guidelines for resubmitting your figure files are available below mine and the reviewer comments at the end of this letter.

We look forward to receiving your revised manuscript.

Kind regards,

Gareth Hagger-Johnson

Academic Editor

PLOS Mental Health

Journal Requirements:

Additional Editor Comments (if provided):

Both reviewers offered a favourable opinion but make some suggestions for strengthening the manuscript from their perspective as practitioners.

My comments mainly focus on the quantitative methods and results.

The analysis is far too complicated, and should be simplified. The research question is not precise enough, which gives the paper a very exploratory feel. I think the paper covers too much ground. The study was not pre-registered, compounding this feeling further. PLOS recommends that studies and research question are pre-registered where possible. https://plos.org/open-science/preregistration/

In the text, the status of covariates/confounders, mediators, effect modifiers (moderators) and outcomes is sometimes obscure (see https://doi.org/10.1093/ije/dyi074). Effect modification is not typically used to test ‘robustness’, it is usually driven by a priori hypotheses about why a predictor or risk factor might have different effects for different groups (‘different slopes for different folks’). Papers usually consider one or two effect modifiers maximum (e.g. age, sex), not four, and I’m not convinced multiple group SEM is the way to do this. Moderated mediation analysis is a possible alternative, but might make things even more complex.

You controlled for age (e.g. Table 1) but not other covariates. I recommend testing for effect modification and if there is no statistical evidence for this, age, sex, work modality etc. should be treated simply as covariates. Include them all in the model and in Table 1, but no need to show them in the path diagram. I had to read the text, figure and table several times in order to work out that status of each variable, which means readers would too, so it needs rewriting to help readers understand clearly what is happening without having to re-read. However, I think this can all be addressed with clearer language, careful introduction of the status of each variable, and a simplified model.

You did not measure a key confounding factor (trait neuroticism) which could be driving most of the covariance over time. It is likely that participants low in a general tendency to experience negative emotions were buffered. It is an unmeasured confounder and an important limitation.

Another limitation is that there are only two timepoints. With three, latent growth curve modelling would have provided a flexible way to handle individual differences and genuine change over time, e.g. parallel process growth models. Ease of interpretation is further complicated by referring to ‘change’ (p.7) in a variable as a mediator. The meaning of ‘change’ is obscure. Is this time 2 value with baseline adjustment for time 2? Or simply the value at time 2? Or the difference between time 1 and time 2? When you only have two timepoints, the meaning of change is not straightforward (see for example https://academic.oup.com/ije/article/51/5/1604/6294759 and the Glymour paper they mention). I think this would be more easily dealt with by focusing on one outcome separately in linear regressions, with cumulative exposure scores for predictors at time 1 and 2 (see below).

If the research question is ‘does emotion regulation cause psychopathology or vice versa’ (this was clearer on p. 21) then please refocus the paper on that question only, keeping the model as simple as possible. Unless there is strong evidence for effect modification, no need to do multiple group analysis, just state where it was not found. Adjust for covariates but show the X, M, Y only at each timepoint in a simplified figure. If you really want to use path models, consider the random-effects cross-lagged panel model (RE-CLPM e.g. https://doi.org/10.3758/s13428-017-0979-2) clearly specifying which is your predictor, mediator and outcome. Regress everything on age, sex, work modality etc. but don’t show these in the path diagram. Only separate groups if effects are materially different, and conclusions. Keep the path diagram as simple as possible (for example, Figure 1 in Deary et al. 2009, Psychology & Aging) and with wide readership in mind.

Even better would be to avoid SEM altogether and estimate mediation using simple linear regression. Exclude any psychopathology at baseline and treat time 1 & 2 emotion regulation (take the mean as a score representing cumulative exposure) as the predictor, time 2 ‘new’ psychopathology as the outcome, everything else as a covariate (except mediators). To test mediation, calculate the % reduction in the X-Y coefficient when each possible mediator is added to the model. Then repeat in a second model, excluding low emotion regulation at baseline, treating time 1 & 2 psychopathology as the predictor, time 2 emotion regulation as the outcome. This should reveal the main direction of possible causation or reverse causation, predictors and mediators of ‘new’ psychopathology and ‘new’ emotion dysregulation (incidentally, this would also mitigate my concerns about unmeasured neuroticism already present at baseline). The ‘exposure’ is time varying hence a ‘cumulative exposure’ (taking the average as a score means always high = highest score, always low = lowest score, everything else scores in between). This would be much easier to understand and reach a wider audience than a complex path model would. You would be able to say what % of the association was mediated/explained, and point to direction of effects, without the complexity in the current version of the manuscript.

In summary, there are several predictors, several mediators, several effect modifiers, and several outcomes – this is all too complex. Please rewrite the introduction to identify a specific gap in the literature, and then introduce a simplified parsimonious model with one (or two) mediating pathways that would answer the question about the most likely causal / reverse causal direction of effects.

Note that where you have done multiple group analysis, dichotomising variables reduces statistical power and is usually not recommended. Another limitation is that it pre-supposes we know in advance where the appropriate cut point is, which will depend on whether the effect modification is linear or not. Some effects are threshold effects, only appearing in the top tertile or even deciles. Some effects are quadratic. The midpoint might not be the place to cut.

I have concerns about the possibility that those who dropped out at Time 2 differ significantly from those who remained in the study. To address this potential issue, a Bonferroni correction is insufficient in my opinion, because it ignores correlations among predictors which could mask important adjusted differences. Instead, I recommend using logistic regression then inverse probability weighting (IPW) to adjust for non-ignorable dropout. This method will help reduce bias in your main analysis.

To implement IPW:

Logistic Regression: Run a logistic regression to predict dropout status at Time 2 (dropout = 1, not dropped out = 0) using all study variables as predictors.

Calculate Weights: Save the predicted probabilities from this logistic regression, and then calculate the inverse of these probabilities to create the weights.

Weight the Analysis: Use these weights in your main model (available in Mplus and other statistical software usually as a survey weight feature). This will adjust the analysis to account for dropout bias.

Comparison and Reporting: Compare the IPW results with the unweighted analysis to assess potential bias. Additionally, report which variables were significant predictors of dropout in your logistic regression (unless there are none - which is highly unusual)

Application in Other Models: If you opt for simple linear regression, you can also apply these weights using survey-weighted regression techniques.

Nonetheless, I think the manuscript has lots of potential and is generally well-written. It is an important topic, and timely. It is also suitable for PLOS Mental Health readers (but please pre-register studies going forward). For these reasons, I recommend a major revision, to allow the analysis to be simplified and the manuscript rewritten to make it more accessible and for maximum impact. Please consider my comments and those of reviewers.

* Please include a STROBE checklist and page numbers where each item in the checklist is addressed, in a revision *

OTHER MINOR EDITS NEEDED

Cover page - Why isn’t the first author the corresponding author?

Abstract – semicolon should be comma

P5 ‘carry over’ not precise enough

P5 Define key terms at first usage (‘Emotion regulation’)

P6 ‘buffer’ isn’t precise enough a term

P4 ‘depression transpires in all spheres of life’ – not clear what this means

P3 You need (at least in the discussion) to reflect on whether the 25% in private practice might be protected by the additional resources they may have.

P8 What proportion are men/women? Why is this missing? If they are mostly women, acknowledge in the discussion as a limitation, but do control for sex as a covariate. You seem to have sufficient numbers of each to test for multiple group analysis so I’m not clear why it’s not reported as a descriptive statistic here.

P9 There is an ethical approval statement here masked for review, then another ethical approval statement on page 10, unmasked. Please include one unmasked review (the review process is transparent anyway, no need to mask).

P9 Extra space in 4,904 deaths. Both death figures need commas.

P11 How can past research be a paper published in 2022, for your study conducted in 2020/21?

P23 Should be ‘begin to dysregulate’ not ‘begin to be dysregulated’

P24 Not controlling for trait negative effect (neuroticism) is a major limitation and should be acknowledged more strongly here. If you had found incremental validity beyond this trait, it would be a stronger argument to focus on these constructs more. I am not convinced yet that the findings aren’t driven by stable personality traits.

P14 A statement should be included that data and code will be deposited in a repository for open access (or the reason if not).

Reviewers' comments:

Reviewer's Responses to Questions

**Comments to the Author**

1. Does this manuscript meet PLOS Mental Health’s publication criteria? Is the manuscript technically sound, and do the data support the conclusions? The manuscript must describe methodologically and ethically rigorous research with conclusions that are appropriately drawn based on the data presented.

Reviewer #1: Yes

Reviewer #2: Yes

2. Has the statistical analysis been performed appropriately and rigorously?

Reviewer #1: I don't know

Reviewer #2: Yes

3. Have the authors made all data underlying the findings in their manuscript fully available (please refer to the Data Availability Statement at the start of the manuscript PDF file)?

Reviewer #1: Yes

Reviewer #2: Yes

4. Is the manuscript presented in an intelligible fashion and written in standard English?

Reviewer #1: Yes

Reviewer #2: Yes

5. Review Comments to the Author

Reviewer #1: I think this makes an interesting and valuable contribution to the literature relevant to mental health practice.

It is important to recognise and be explicit about the practice context of the research participants, e,g, cultural, professional-clinical and regulatory factors may contribute and also affect application/generalisability of findings.

Modifying factors, such as requirements for the provision and uptake of clinical supervision, caseload/workload restrictions, vary significantly between professions, countries/state regulators of professions and providers of services, and in their evidence base.

It would be insightful to understand if there is significant difference in reported outcomes when demographic and contextual factors are triangulated.

Reviewer #2: The research article was a clear and thorough analysis into the impact that stressors have on emotional regulation have over time. I feel that the decision to take advantage of the coronavirus pandemic was a good one because it allows the ability to provide research under conditions and stressors that would otherwise be unethical.

The paper is very well referenced and I didn't see any areas for improvement in this regard.

The scales used in the study are research based and comparable which adds to the overall strength and validity to the study. In future papers, it would be interesting to see a qualitative of mixed methods approach. This paper is strong in its reliability and replicability however may be reductive to individual experiences in that context and time. This does not reduce the effectiveness of the current paper at all as the goals of the paper are very clear. it meets the goals that it sets out well.

I really like how different professional backgrounds are highlighted and considered. Being a social worker in a mental health team myself, I often feel like a minority and my professional knowledge and experience can be reduced to silence. One of the limitations that the paper highlights was that there was not equal representation of professional groups. This is likely the case due to mental health teams being largely health and psychology lead. An interesting area for future study would be to explore more specifically the variances between different professional groups. Policies that we have had in the UK aimed at creating a more generic professional role in mental health such as “new ways of working” as well as the change from the Approved social worker (ASW) to the Approved Mental Health Practitioner (AMHP) erode professional identity. I’m not familiar with Canadian policies but it is well documented that in a workplace setting lead by a medical institution, some professional identities can be lost due to introduction of generic roles such as care-coordinator or allied health professional. For instance, Nathan and Webber (2010) argued that the specialised knowledge and skills of the social worker (social and environmental determinants of mental health) can be lost to a more bio-medical infrastructure. I would be interested to understand more how the loss of professional identity may impact job satisfaction which may in turn impact compassion fatigue and depression over time. My hypothesis would be that those with a non-medical background such as social work would be less resilient to longer term depression and compassion fatigue due to feeling less satisfied with their contribution and not having the same sense of value.

Another thing that stood out to me was that the overwhelming majority of the sample was female (89.5%). It would be interesting to have a more focused study into how stressors can implicate emotional regulation in male mental health practitioners, or to compare the variations between males and females in more detail. There may be differences t how they regulate or supress emotions based in socially expected behaviour on an individual level as well as wider societal and cultural norms.

The paper discusses clinical implications and offers some examples of of how work conditions may be addressed. One recommendation that I would make would be for government to adequately fund and provide additional resource to mental health teams., which would reduce stressors. A lot of emphasis is placed on the individual when conversations are had about burnout or compassion fatigue. Mental health services are chronically under funded. Mental health professionals are forever being told to take breaks, do self care, and get adequate supervision, but the diversion of the problem onto the individual can only go so far.

I was really impressed with this article. It was a really interesting read which really made me reflect on my own experience as a social worker in a community mental health team, as well as that of the rest of my colleagues.

Nathan, J. and Webber, M. (2010) Mental Health Social Work and the Bureau-Medicalisation of Mental Health Care: Identity in a Changing World. Journal of Social Work Practice. 24(1) 15-28.

6. PLOS authors have the option to publish the peer review history of their article (what does this mean?). If published, this will include your full peer review and any attached files.

**Do you want your identity to be public for this peer review?** For information about this choice, including consent withdrawal, please see our Privacy Policy.

Reviewer #1: No

Reviewer #2: **Yes: **Will Mclennan

---

## [Decision Letter · Decision Letter 1]

25 Nov 2024

Emotion Regulation and Compassion Fatigue in Mental Health Professionals in a Context of Stress: A Longitudinal Study

PMEN-D-24-00323R1

Dear Dr. Philippe,

We are pleased to inform you that your manuscript 'Emotion Regulation and Compassion Fatigue in Mental Health Professionals in a Context of Stress: A Longitudinal Study' has been provisionally accepted for publication in PLOS Mental Health.

Best regards,

Gareth Hagger-Johnson

Academic Editor

PLOS Mental Health

Thank you for this revision, which addresses the reviewers' and my comments and requests.

Reviewer Comments (if any, and for reference):

Reviewer's Responses to Questions

**Comments to the Author**

1. If the authors have adequately addressed your comments raised in a previous round of review and you feel that this manuscript is now acceptable for publication, you may indicate that here to bypass the “Comments to the Author” section, enter your conflict of interest statement in the “Confidential to Editor” section, and submit your "Accept" recommendation.

Reviewer #1: All comments have been addressed

2. Does this manuscript meet PLOS Mental Health’s publication criteria? Is the manuscript technically sound, and do the data support the conclusions? The manuscript must describe methodologically and ethically rigorous research with conclusions that are appropriately drawn based on the data presented.

Reviewer #1: Yes

3. Has the statistical analysis been performed appropriately and rigorously?

Reviewer #1: I don't know

4. Have the authors made all data underlying the findings in their manuscript fully available (please refer to the Data Availability Statement at the start of the manuscript PDF file)?

Reviewer #1: Yes

5. Is the manuscript presented in an intelligible fashion and written in standard English?

Reviewer #1: Yes

6. Review Comments to the Author

Reviewer #1: (No Response)

7. PLOS authors have the option to publish the peer review history of their article (what does this mean?). If published, this will include your full peer review and any attached files.

**Do you want your identity to be public for this peer review?** For information about this choice, including consent withdrawal, please see our Privacy Policy.

Reviewer #1: No
